# Inhibiting the reproduction of SARS-CoV-2 through perturbations in human lung cell metabolic network

Hadrien Delattre[1], Kalesh Sasidharan[1,2] , Orkun S Soyer[1,2]

**Viruses rely on their host for reproduction. Here, we made use of genomic and structural information to create a biomass function capturing the amino and nucleic acid requirements of SARS-CoV-2. Incorporating this biomass function into a stoichiometric metabolic model of the human lung cell and applying metabolic flux balance analysis, we identified host-based metabolic perturbations inhibiting SARS-CoV-2 reproduction. Our results highlight reactions in the central metabolism, as well as amino acid and nucleotide biosynthesis pathways. By incorporating host cellular maintenance into the model based on available protein expression data from human lung cells, we find that only few of these metabolic perturbations are able to selectively inhibit virus reproduction. Some of the catalysing enzymes of such reactions have demonstrated interactions with existing drugs, which can be used for experimental testing of the presented predictions using gene knockouts and RNA interference techniques. In summary, the developed computational approach offers a platform for rapid, experimentally testable generation of drug predictions against existing and emerging viruses based on their biomass requirements.**

## Introduction

One of the crucial steps in the virus life cycle is the synthesis of the virus particles within the host cell. This includes the synthesis of viral structural proteins and new genomic material. For these processes, all viruses are fully dependent on their host for the required energy (Mahmoudabadi et al, 2017) and building blocks (Berzin et al, 1974; Waldbauer et al, 2019). This dependency is evidenced by experimental findings showing significant metabolic flux alterations in host cells upon infection (Maynard et al, 2010a; Yu et al, 2011). System-level metabolic studies have particularly highlighted changes in glucose uptake and glycolysis (El-Bacha et al, 2004; Munger et al, 2006), which might be related to an increased demand for biosynthetic precursors as viral production becomes the dominant process within infected cells (Berzin et al, 1974).

The observation of virus synthesis dominating the metabolism and physiology of infected cells suggests that it might be possible to manipulate cell metabolism to control the viral infection (Ikeda & Kato, 2007; Maynard et al, 2010). Indeed, several of the existing antivirals such as ribavirin, remdesivir, and gemcitabine are nucleoside analogs that target metabolic enzymes in the nucleotide biosynthesis pathways and are thought to function through their impact on free nucleotide pools in the cell (Leyssen et al, 2008; Wang et al, 2011). An even more specific metabolic approach to inhibit virus production was demonstrated in the case of human cytomegalovirus. For this virus, metabolic analyses highlighted a shifting of metabolic fluxes within central carbon metabolism and fatty acid biosynthesis pathways during infection (Munger et al, 2008). It was predicted that these flux changes could be blocked by perturbation of specific enzymes, which were then targeted with available inhibitors and resulted in reduced virus production (Munger et al, 2008). Systematic analysis of gene knockout effects on infection of bacteria with phage also identified many metabolic genes associated with central carbon metabolism and substrate transport (Maynard et al, 2010b), leading to the proposition of using host metabolic engineering to modulate viral production (Maynard et al, 2010a). Such metabolic control has been explored in virus-based bioproduction using insect cells, where alterations in the culture media allowed alteration of metabolic fluxes and production levels (Carinhas et al, 2010).

These experimental findings show that viral biomass synthesis causes significant metabolic flux changes in host cells and that metabolic perturbations can directly alter virus reproduction. Thus, system-level metabolic models could be used to predict what types of metabolic alterations can cause what kinds of impact on virus reproduction. Although modelling of virus reproduction in host cells has mostly taken a kinetic approach, focusing on translation and transcription processes (Endy et al, 1997; You et al, 2002; Yin & Redovich, 2018), it has been possible to combine such kinetic models with genome-scale metabolic models to account for both host and virus biomass (Jain & Srivastava, 2009). This allowed predicting the effects of metabolites available in the culture media on the dynamics of the infection process (Birch et al, 2012). In a human cell context, genome-scale metabolic models were used to analyse the metabolic impact of infection of macrophages with

---

[1]School of Life Sciences, University of Warwick, UK   [2]Bio-Electrical Engineering Innovation Hub, University of Warwick, UK

Correspondence: O.Soyer@warwick.ac.uk

bacteria or viruses (Bordbar et al, 2010; Aller et al, 2018). One of these studies incorporated viral production into the macrophage metabolic model and predicted specific reaction perturbations that can cause a reduction in viral reproduction (Aller et al, 2018). These predictions correctly identified enzyme targets of the aforementioned antiviral drugs in nucleotide pathways and highlighted new target enzymes (Aller et al, 2018). Such findings from the virus-host metabolic modelling aligns with the observations that genome-scale metabolic models can provide a comprehensive stoichiometric catalogue of possible biochemical conversions in a cell (Thiele et al, 2013; Swainston et al, 2016) and can generate useful qualitative predictions on the impact of environmental or genetic alterations on the cellular metabolic flux distributions (Edwards & Palsson, 2000; Segrè et al, 2002; Papp et al, 2004).

Motivated by the qualitative predictive power of stoichiometric metabolic models and flux analysis, we apply it here to simulate the production of SARS-CoV-2 virus particles as part of the host metabolism and predict metabolic inhibitions against this virus. Given the emerging literature on SARS-CoV-2 infection primarily targeting lung and intestinal cells, we focus here on human lung cells as the host system. We adapt the available human genome-scale metabolic model with a biomass maintenance function based on human lung tissue–derived expression data, and include in this model also a viral biomass reaction, estimated using structural information available from SARS-CoV-2 and related viruses. By optimising flux distributions in this "infected lung cell model" for viral biomass reaction, we were able to predict reactions whose suppression or constraint can theoretically inhibit viral reproduction. We explored the possible impact of these predicted inhibitions on the host metabolism itself, as well as the experimental feasibility of implementing the predicted metabolic perturbations using available drug and inhibitor information on metabolic enzymes. Our results indicate that individual and double perturbation of several metabolic reactions from central metabolic pathways can inhibit SARS-CoV-2 reproduction in general and some of these can do so selectively without affecting normal metabolic functions of the host. We highlight these reactions as experimentally testable drug targets for inhibiting SARS-CoV-2 reproduction in human lung cells and provide details of the implemented computational approach for further development.

## Results

To simulate the production of SARS-CoV-2 virus particles in a human cell, we used an existing, community-developed human genome scale model known as RECON2.2 (Thiele et al, 2013; Swainston et al, 2016) (see the Materials and Methods section). This model represents the most comprehensive catalogue of metabolic reactions found in human cells, with many of its reactions associated with known genes (Swainston et al, 2016). Within this model, we implement a pseudo reaction representing the production of SARS-CoV-2 viral particles from biosynthetic precursors (see Fig 1A and the Materials and Methods section). The construction of this pseudo reaction is based on available structural information on the virus including its use of proteins for viral packaging (Neuman et al,

2006; 2011; 2008; Bárcena et al, 2009, Mahmoudabadi et al, 2017, Bar-On et al, 2020) and its genome sequence. As such, this pseudo biomass reaction accounts for the stoichiometry of nucleic and amino acids required to make a complete virus and associated energetic costs. This analysis highlights that leucine and alanine are the most used amino acids in SARS-CoV-2 proteins, and adenosine- and uridine-triphosphate are the more common nucleotides in its RNA (Fig 1B).

### Metabolic fluxes supporting SARS-CoV-2 production in a human cell are primarily in central carbon metabolism

By incorporating the SARS-CoV-2 viral biomass function into the human metabolic model and assuming a minimal media composition (see the Materials and Methods section), we predict a metabolic flux distribution for optimal virus production in a human cell (Fig 1C). We then evaluated the flux variability allowed in each reaction of the model, while maintaining an optimal viral production level (Table S1). These analyses have shown that reactions which must carry flux for optimal viral biomass production include glycolysis, oxidative phosphorylation, fatty acid oxidation, and specific amino and nucleic acid biosynthesis pathways (Fig 1C and Table S1). As the optimal flux distributions are related also to the flux limits imposed on uptake reaction fluxes, we also repeated the flux variability analysis with minimal media but using an increased uptake limit, and with a rich media that allows all uptake reactions of the model to be active. Increasing the uptake limits did not alter the general conclusions about key active pathways sustaining optimal virus production but resulted in higher uptake fluxes, which cause additional pathways relating to overflow metabolism to be active (Table S1). Simulating a rich media resulted in a much lower number of flux-carrying reactions, as the cell can obtain several key compounds such as uridine triphosphate from the media under this scenario (Table S1). Because this rich media allows all transport reactions in the RECON2.2. model to carry flux, providing the cell access to most building blocks, we believe it is physiologically rarely, if it all, achieved (e.g., limitation of amino acids). We therefore focus the remaining analysis on the results from simulations using the minimal media.

### Inhibiting specific metabolic enzymes and enzyme combinations inhibit SARS-CoV-2 production in human cells

To identify reaction perturbations which, when inhibited, can halt or reduce virus production, we systematically simulated a knockout of each flux-carrying reaction. Excluding reactions involved in uptake from media, this analysis highlighted 35 reaction knockouts that can stop virus production and an additional eight reactions that can reduce it below 80% of the original (Fig 1C). The former group of reactions tended to be involved in nucleotide and amino acid biosynthesis pathways, whereas the latter group included reactions primarily in glycolysis and oxidative phosphorylation (Tables 1 and S2). Key ones among these reactions are further discussed below.

Considering that it is possible for the effects of single perturbations to be circumvented by re-directing of fluxes, we also

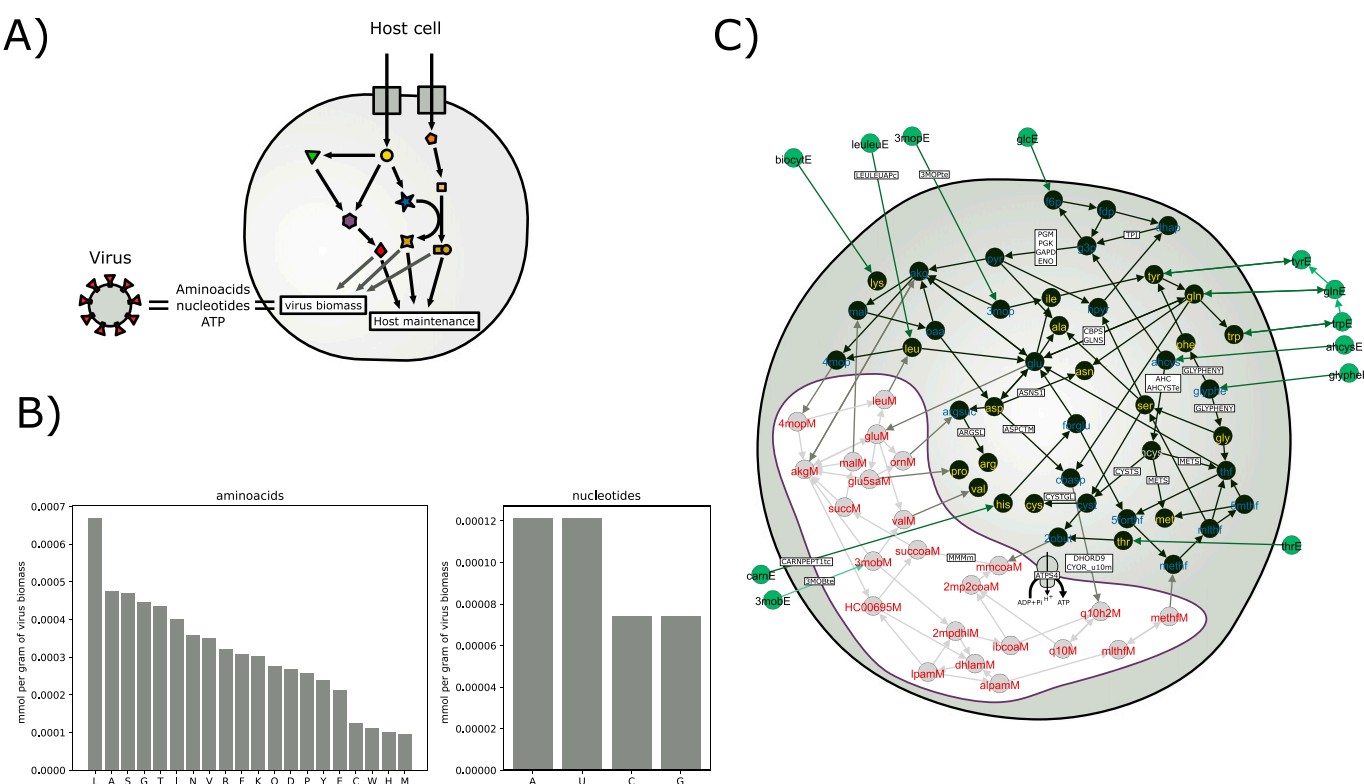

**Figure 1. Model schematic, virus biomass analysis, and model fluxes. (A)** Schematic representation of the integrated host–virus metabolic modelling approach used in this article. The biomass composition of SARS-CoV-2 is estimated as described in the Materials and Methods section and then embedded in the metabolic network model of the host cell. This model is then used to predict the metabolic fluxes supporting virus production and effects of perturbations as described in the main text. **(B)** Composition of virus biomass in mmol per gram of virus biomass dry weight. The two panels show amino acids and nucleotides as labelled. **(C)** Graph representation of part of the human cellular metabolic network, focusing on those reactions that are active in virus production under minimal media conditions with uptake fluxes set to −10 mmol gDW$^{-1}$ h$^{-1}$ (see main text and the Materials and Methods section). The cell is shown as a circle with a grey background, with the mitochondrial matrix shown in white background. Nodes are metabolites and edges are reactions. Edges between two nodes are drawn when at least one reaction connecting those two metabolites carries flux in the optimal flux distribution. The graph shown here is obtained from the full metabolic network by selecting those edges corresponding to the shortest path between metabolites present in the medium and the precursor metabolites to amino acids that contribute to the virus biomass. The paths are found by using the Dijkstra algorithm (Dijkstra, 1959) and weighting the edges by the inverse of their flux value. Nodes are coloured by their cellular location; cytosol (black), mitochondrial matrix (grey), and external medium (green). Node labels are coloured also according to primarily location, with amino acids and media components labelled in green and yellow, respectively. Key reactions discussed in the text and summary results are labelled on their corresponding edges.

explored combined perturbations. We created all possible pairs of flux-carrying reactions according to the flux variability analysis (more than 5,000 pairs) and simulated the effect of setting their reaction fluxes to zero. This has identified more than 400 reaction pairs, co-inhibition of which results in the reduction of virus optima to 80% or less of the original (Tables 1 and S2). Most of these reaction pairs involved one of the 10 single perturbations that were found to reduce virus production to less than 80% on their own, but pairing them with additional reaction increased their impact. For example, inhibition of GAPD and cytochrome c oxidase (CYOR) individually caused reduction to 62% and 60% of original virus production respectively, but combined inhibition of these reactions results in 25% of original production (Table S2). Some of these cases of increased effect arise due to co-inhibition of reactions more effectively blocking fluxes into virus biomass precursors. For example, combined blocking of GAPD and CYOR, reactions involved in respiration and glycolysis, respectively, results in reduced fluxes into pyruvate and alpha-ketoglutarate (akg), a key intermediary of the tricarboxylic acid (TCA) cycle. Akg is further linked into valine

production through a valine:3-methyl-2-oxobutanoate shuttle across the mitochondrial membrane (Fig 2). In the optimal flux distribution for SARS-CoV-2 production, this "valine shuttle" has a high flux and contributes to the production of both valine and multiple other amino acids via mitochondrial glutamate (Fig 2B). Perturbations to both CYOR and GAPD lead to a new flux distribution where the glutamate production in the mitochondrial matrix is sustained through a different metabolic route (Fig 2A). The "valine shuttle" that was active in the optimal solution is now non-functional and is instead replaced by a leucine: 4-methyl-2-oxopentanoate shuttle carrying a lesser flux. This in turn decreases the production of the amino acids from glutamate, and thus causing a significant decrease in virus biomass production flux (Fig 2).

In the case of simulating the minimal media with higher uptake fluxes, we have also identified pairs of completely new enzyme inhibitions, which were not causing any effect on their own (Table S2). Some of these pairs exert their effects by blocking multiple pathways from a given compound and thereby causing disruption

**Table 1.   Selection of reaction perturbations predicted to reduce SARS-CoV-19 biomass production in a human cell with equal or less impact on human lung cell-based metabolic maintenance.**

| Reaction 1 | Reaction 2 | Virus optima | Host optima | Perturbation |
|---|---|---|---|---|
| ATPS4m | — | 75 | 77 | Knockout |
| ENO | — | 65 | 66 | Knockout |
| PGM | — | 66 | 67 | Knockout |
| CYOR | — | 61 | 69 | Knockout |
| CYOR | GAPD | 25 | 30 | Knockout |
| CYOR | PUNP3 | 60 | 69 | Knockout |
| CYOR | ASPTA | 60 | 68 | Knockout |
| PGK | FTCD | 58 | 59 | Knockout |
| ASPCTr | — | 76 | 100 | Enforcement |
| CBPS | — | 76 | 100 | Enforcement |
| DHORD9 | — | 76 | 100 | Enforcement |
| DHORTS | — | 76 | 100 | Enforcement |
| OMPDC | — | 76 | 100 | Enforcement |
| ORPT | — | 76 | 100 | Enforcement |
| ASNS1 | — | 78 | 100 | Enforcement |
| GLNS | — | 84 | 100 | Enforcement |
| THRD_L | — | 84 | 100 | Enforcement |
| LEULEULAPc | — | 90 | 100 | Enforcement |
| LEULEUPEPT1tc | — | 90 | 100 | Enforcement |
| ASPCTr | DHFR | 73 | 100 | Enforcement |
| CBPS | DHFR | 73 | 100 | Enforcement |
| DHORD9 | DHFR | 73 | 100 | Enforcement |
| DHORTS | DHFR | 73 | 100 | Enforcement |
| OMPDC | DHFR | 73 | 100 | Enforcement |
| ORPT | DHFR | 73 | 100 | Enforcement |

Individual or pairs of reaction perturbations are shown, alongside their predicted effects on SARS-CoV-19 and host as percent of optima without any perturbations. For full results, see Tables S2 and S5. Reactions are identified with the short notation used in the RECON2.2 model and their gene and subsystem associations are given in Table S1. Short notations used are: ATPS4m, ATP synthase; ENO, enolase; PGM, phosphoglycerate mutase; CYOR, ubiquinol-6 cytochrome c reductase; GAPD, glyceraldehyde-3-phosphate dehydrogenase; PUNP3, purine-nucleoside phosphorylase (Guanosine); ASPTA, aspartate transaminase; PGK, phosphoglycerate kinase; ASPCTr, carbamoyltransferase; CBPS, carbamoyl-phosphate synthase; DHORD9, dihydroorotic acid dehydrogenase; DHRTS, dihydroorotase; OMPDC, orotidine-5-phosphate decarboxylase; ORPT, orotate phosphoribosyltransferase; ASNS1, asparagine synthase; GLNS, glutamine synthase; THRD, threonine deaminase.

in steady state balances in the model. For example, co-inhibition of citrate synthase (CSm) and several other enzymes such as histidase (HISD) totally prevents flux in SARS-CoV-2 biosynthesis reaction by making impossible the mass balance of protons in the cytosol (see Fig S1).

## Metabolic requirements of viral production are different to those arising from host cellular maintenance

In the above discussed analyses, we considered host metabolism as optimised for viral production and evaluated impact of perturbations only on this process. Such metabolic perturbations should also be evaluated for their impact on the normal metabolic functions of uninfected host cells. In previous studies, normal state of metabolism in human cell lines are either represented through a

pseudo reaction for cellular maintenance (Bordbar et al, 2010; Thiele et al, 2013) or through consideration of specific metabolic functions such as ATP or lipid production (Mardinoglu et al, 2014). In the former case, cell maintenance is captured by a generic account of cellular constituents such as lipids, carbohydrates and DNA and a more specific accounting of amino acid usage in protein expression (Bordbar et al, 2010; Thiele et al, 2013). In RECON2.2, the protein-based component of the maintenance function is calculated from a large collection of human genes' open reading frames (Thiele et al, 2013).

Here, we expanded from this approach to focus on human lung cells, which are shown to be the primary target of SARS-CoV-2 infections along with intestinal cells (Cagno, 2020). To create a lung cell–specific biomass maintenance function, we calculated the biomass protein components using available expression data

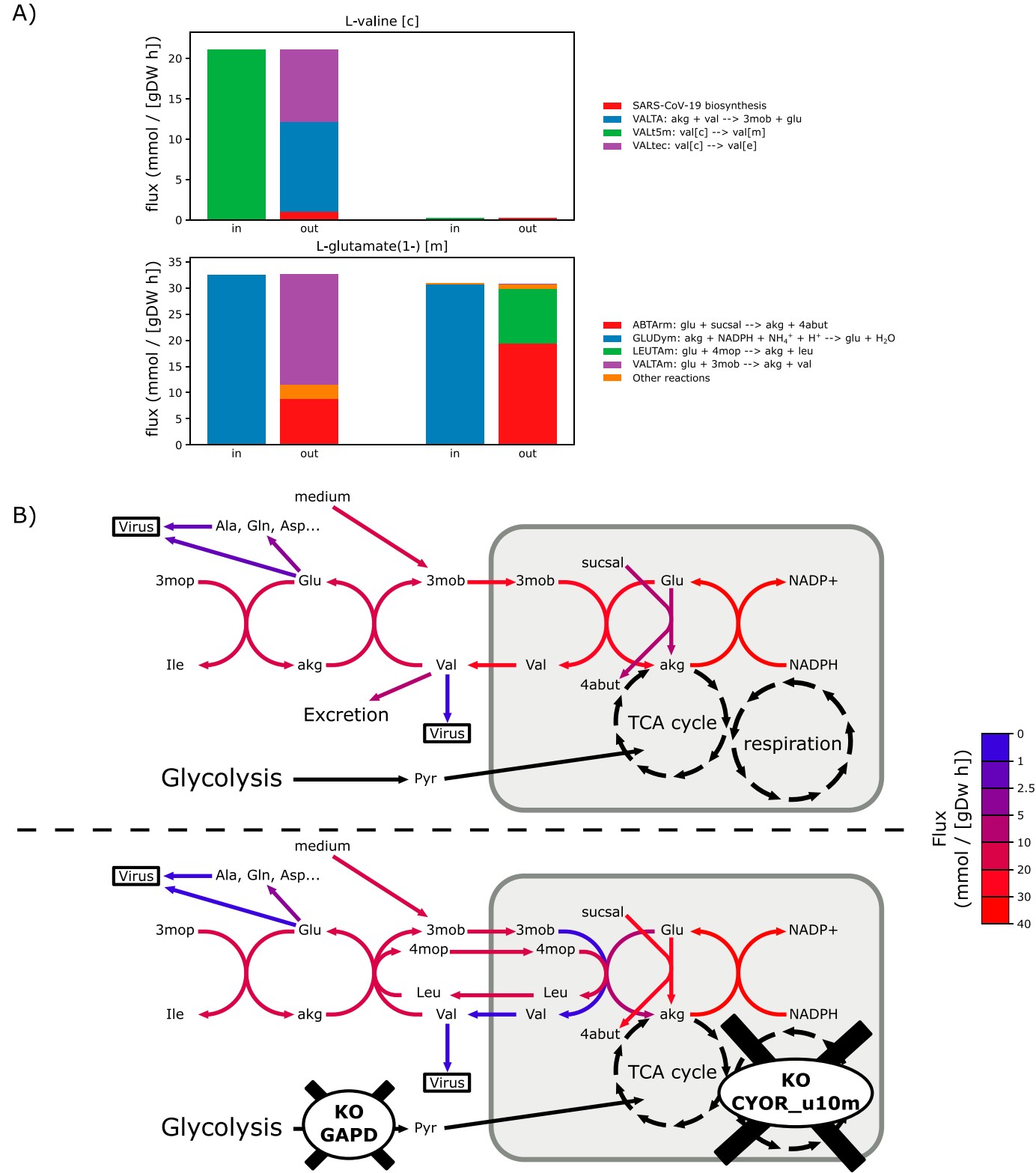

**Figure 2. Effect of the double knockout of GAPD and cytochrome c oxidase (CYOR) on the optimal flux distribution for virus production under minimal media conditions with uptake fluxes set to −10 mmol gDW−1 h−1.**

**(A)** Distribution of the producing ("in") and consuming ("out") fluxes for valine in the cytosol and glutamate in the mitochondrial matrix. For each metabolite, the in and out fluxes have been computed for the unmodified model (left) and for the double knockout of GAPD and CYOR (right). **(B)** Cartoon representation of the reaction network involving valine and other metabolites, with reaction fluxes from normal and perturbation conditions colour-mapped onto reaction arrows. The upper panel represents

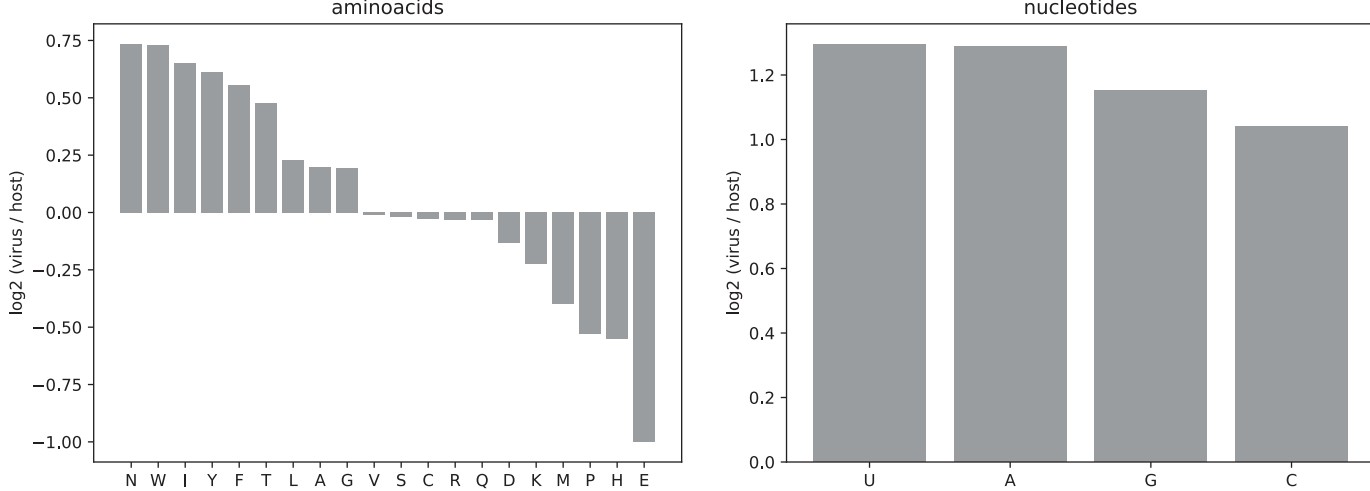

**Figure 3.  Compositional comparison of SARS-CoV-2 biomass and host maintenance based on human lung cell protein expression.**
The values on the y-axis of both panels are the base 2 logarithms of the ratio of the proportion of each metabolite (amino acid or nucleotide) in the virus biomass divided by its proportion in the host maintenance function. The proportion of each metabolite is its stoichiometric coefficient in the corresponding reaction (virus biomass or host maintenance) in mmol gDW$^{-1}$, divided by the sum of all stoichiometric coefficients in that reaction.

available from the Human Protein Atlas project for lung tissue (Uhlén et al, 2015) (see the Materials and Methods section). Comparing the resulting human lung cell maintenance function to the SARS-CoV-2 biomass, in terms of the building block stoichiometries, revealed differences in relative amino and nucleic acid usage (Fig 3 and Table S3). Compared with the host, there was particularly higher relative usage of phenylalanine, isoleucine, asparagine, threonine, tryptophan, and tyrosine in the virus and particularly lower relative usage of glutamate, histidine, methionine, and proline (Fig 3). Accordingly, the optimisation of the model using the host metabolic maintenance results in a different metabolic flux distribution compared with viral production (Table S4). The differences, however, were rather limited from the perspective of fluxes supporting SARS-CoV-2 production; out of all reactions that must carry a flux to sustain a virus optimal state, almost all were also required to carry flux to sustain a host optimal state (Table S4).

**Flux control can ensure selective reduction in viral production**

Given the above finding that the same reactions carrying flux for SARS-CoV-2 production also carry flux for host metabolic maintenance, we re-analysed the effects of enzyme perturbations on both virus and the host. We found that many of the previously identified single perturbations limiting virus production also limited significantly the host metabolic maintenance, with only one single perturbation—that involving CYOR—showing more than 5% difference in its impact on virus versus the host (Table S5). The same finding prevailed for double perturbations. The only pairs that displayed 5% or more difference in their effects on virus versus host are those involving CYOR paired with other enzymes (Table S5).

The limited differential impact of full inhibition of enzymes made us postulate that more refined perturbations could provide a better strategy to just impact SARS-CoV-2 production without affecting the host. In particular, given the differences in optimal metabolic fluxes between virus production and host maintenance states, we argued that there might be flux values for some reaction that are compatible with only one of these states. To explore this possibility, we systematically analysed the flux variability of each reaction given either the optimisation of host maintenance or virus production. This has allowed us to see if any of the reactions would have flux regimes that are only compatible with the optimal host maintenance but not with optimal virus production and then "enforce" such flux regimes on the model. This approach allowed us to identify few single and double reaction perturbations that are fully selective on their effect and solely reduce virus production without causing any impact on the host (see Table S6). They involved a small number of metabolic reactions, namely, aspartate carbamoyltransferase (ASPCTr), carbamoyl-phosphate synthase (CBPS), dihydroorotic acid dehydrogenase (DHORD9), dihydroorotase (DHRTS), orotidine-5-phosphate decarboxylase (OMPDC), orotate phosphoribosyltransferase (ORPT), asparagine synthase (ASNS1), glutamine synthase (GLNS), and threonine deaminase (THRD) and caused up to 27% reduction in SARS-CoV-2 production (Table S6).

The flux enforcement approach creates further constraints on how the metabolic fluxes in the system can be balanced at steady state. For example, in the optimal flux distribution for SARS-CoV-2 production in the unmodified model, threonine is obtained from the medium through a threonine:leucine shuttle, with both of these amino acids being in relatively similar demand between host and virus requirements. In the case of THRD fluxes enforced to specific

the flux distribution in the normal condition, whereas the lower panel represents flux distribution under perturbation, that is, when both GAPD and CYOR are knocked out. The light grey rectangle represents the mitochondrial compartment. Metabolite notations used are: 3-methyl-2-oxobutanoate (3mob), 4-methyl-2-oxopentanoate (4mop), alpha-ketoglutarate (akg), 4-oxobutanoate (sucsal), and gamma-aminobutyric acid (4abut).

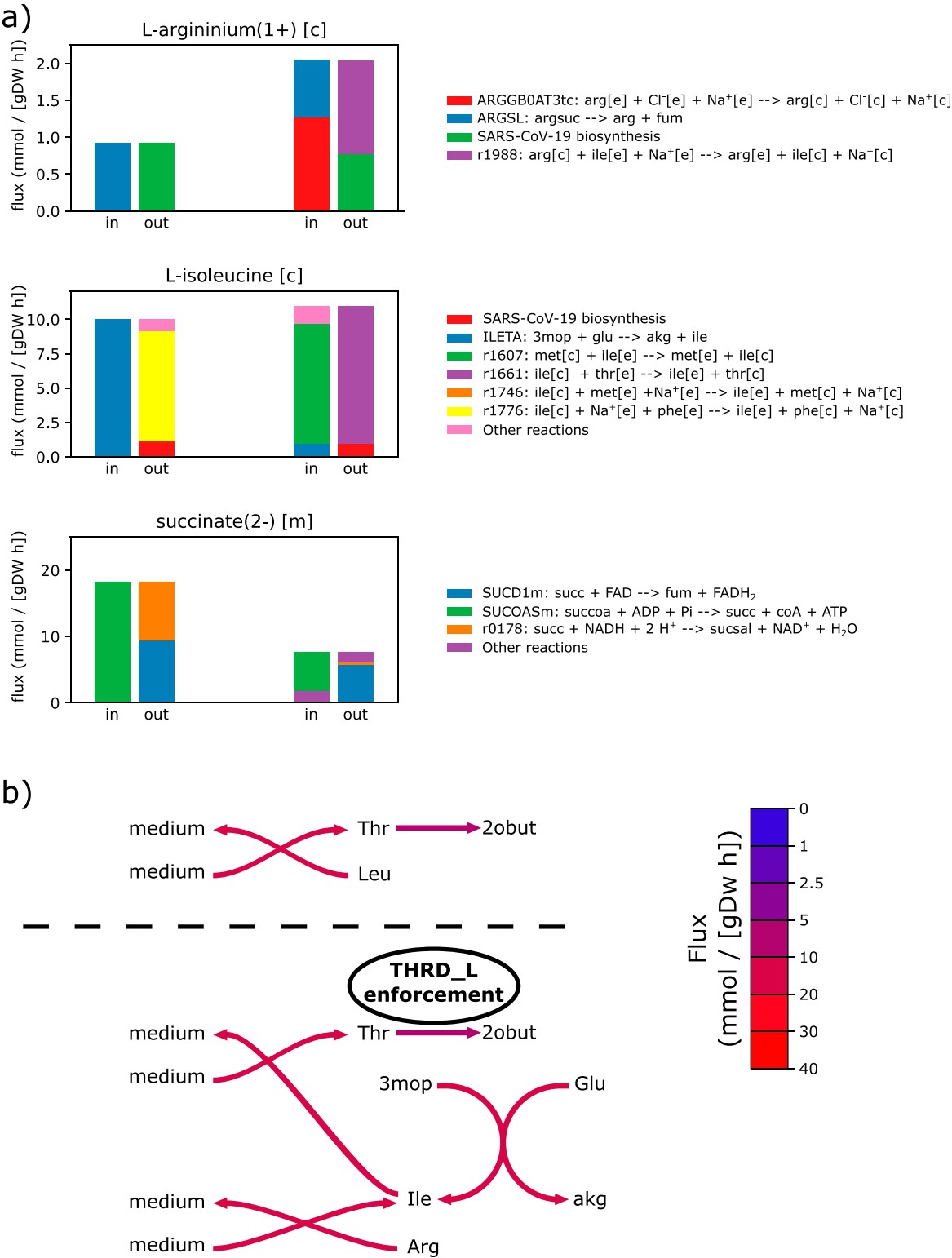

**Figure 4.  Effect of the flux enforcement of threonine deaminase (THRD) on the optimal flux distribution for SARS-CoV-2 production.**
The enforced flux boundaries were (8.94; 9.14) mmol gDW$^{-1}$ h$^{-1}$ for THRD. **(A)** Distribution of the producing ("in") and consuming ("out") fluxes for arginine and isoleucine in the cytosol and succinate in the mitochondrial matrix. For each metabolite, the in and out fluxes have been computed in two different conditions; from left to right; unmodified model and flux enforcement of THRD only. **(B)** Cartoon representation of the reaction network involving threonine and other amino acids, with reaction fluxes from normal and perturbation conditions shown above and below the dashed line, respectively. Flux values are colour-mapped onto reaction arrows. Metabolite

Table 2.  Existing drugs that are approved or investigated for clinical use and that can target some of the enzymes predicted in this study as causing metabolism-based reduction of SARS-CoV-2 biomass production.

| Reaction | Enzyme | Approved drug or inhibitor | Current use | DrugBank ID | References (PubMed ID) |
|----------|--------|---------------------------|-------------|-------------|------------------------|
| ATPS4m | ATP synthase subunit a | Estradiol | Hormone therapy | DB00783 | 12587531; 23274738 |
| ATPS4m | ATP synthase subunit c | Bedaquiline | Antimicrobial | DB08903 | 17496888 |
| ENO | Enolase | Sodium fluoride | Antimicrobial | DB09325 | 9227132; 16411755 |
| PGM/PGK | Phosphoglycerate mutase | Artenimol | Antimicrobial | DB1163 | 26340163 |
| PGM/PGK | Phosphoglycerate mutase | Copper | Diet supplement | DB09130 | 15359738 |
| FTCD | Formimidoyltransferase cyclodeaminase | Pyridoxal phosphate | Diet supplement | DB00114 | 17016423 |
| SUCD1m | Succinate dehydrogenase | Ubidecarenone | Diet supplement | DB09270 | 16551570 |

Information is collated from DrugBank database (Wishart et al, 2018). The reaction abbreviations are as in Table 1.

ranges, threonine is instead obtained through a threonine:iso-leucine shuttle and is now further intertwined with the arginine:isoleucine shuttle, thereby creating a trade-off among these amino acids (Fig 4). Crucially, arginine is the highest and isoleucine the third highest differentially demanded amino acid when comparing virus with host requirements (Fig 3). This is why the described trade-off situation among these amino acids and threonine, caused by the flux enforcement, differentially impacts the SARS-CoV-2 bio-mass production more than it effects host maintenance.

Motivated by this finding, we also simulated the effect of the decreased availability of specific metabolites in the culture medium on the host and virus biomass fluxes. The maximum import rate of the different molecules of the minimal medium we defined was set to values ranging between 0.1 and 10 mmol/gDW/h (where DW stands for grams of dry weight) and host and viral biomass optima compared (Table S7). As the maximum import rate of any molecule of the minimal medium decreases, so does the value of the optimal biomass flux of both host and virus. In some cases, namely, for threonine, glycylphenylalanine, and 3-methyl-2-oxovalerate, this decrease occurs more readily for the virus' optimal biomass flux than for the host. This could be interpreted that a situation of low availability of the aforementioned molecules in the medium is more detrimental to the virus replication than to the host's maintenance.

### Current metabolic drugs exist that could target predicted reactions to inhibit production of SARS-CoV-2

The metabolic approach used here allowed prediction of several reactions and reaction combinations that could limit SARS-CoV-2 production in human cells in general and differentially in human lung cells. The significant ones of these are re-summarised in Table 1 as those reactions, the perturbation of which, can reduce virus production below 80% of the original (see Supplementary files for full results). For these reactions, we evaluated their associated enzymes in the light of existing, approved drugs using the available small molecule inhibitor and drug database DrugBank (Wishart et al, 2018). We found several existing drugs that could inhibit some

of the predicted reactions including those targeting enolase (ENO), phosphoglycerate mutase (PGM), and SUCD1m (Table 2). These drugs could be used as a starting point to experimentally test the predictions made here, using in vitro assays. In addition to these identified small molecule inhibitors, we note that it might also be possible to achieve development of de novo metabolic gene knockout approaches using recent CRISPR and RNA-silencing approaches.

## Discussion

Here, we have created a stoichiometric biomass function for the COVID-19–causing SARS-CoV-2 virus and incorporated this into a human lung cell genome scale metabolic model. The viral biomass function highlights the key building blocks required to synthesize a SARS-CoV-2 virus and its simulation within the human metabolic model enables predicting optimal flux distributions in the host for sustaining either SARS-CoV-2 reproduction or host maintenance. We used this capability to identify reaction perturbations that can inhibit SARS-CoV-2 reproduction in general or selectively, without inhibiting the host metabolic maintenance. The identified reactions primarily fall onto glycolysis and oxidative phosphorylation pathways and their connections to amino acid biosynthesis pathways. The latter finding is in line with the additional observation we made here that the relative stoichiometries of specific amino acids differ in SARS-CoV-2 biomass versus host cell maintenance estimated using human lung cell protein expression data. Together, these results highlight the possibility of targeting host metabolism for inhibition of SARS-CoV-2 reproduction in human cells in general and in human lung cells specifically.

The predictions presented here are based crucially on the structure of the metabolic model as well as the two key assumptions of the flux balance analysis (FBA), namely, the assumptions of metabolic steady state and the optimality of metabolic fluxes towards a specific metabolic function. In the case of the key assumptions of FBA (Schuster et al, 2008; Raman & Chandra, 2009), these are expected not to affect qualitative predictions

notations used are: threonine (Thr), 2-oxobutanoate (2obut), leucine (Leu), isoleucine (Ile), arginine (Arg), glutamate (Glu), 3-methyl-2-oxovalarate (3mop), and alpha-ketoglutarate (akg).

on which metabolic reactions might be required to carry flux for a given metabolic process or how specific perturbations might impact such processes. For example, FBA-based approaches have been successful in predicting and explaining experimental observations on gene deletion and environmental perturbations in both microbial (Ibarra et al, 2002) and eukaryotic systems (Papp et al, 2004). Ultimately, the "infected host cell models" should be developed in conjunction with experiments on infected cells, as well as detailed information on virus stoichiometry, which has been performed in the case of other viruses (Munger et al, 2006).

There is unfortunately not much available literature yet on SARS-CoV-2–infected cells and their metabolism, to allow us detailed comparison between model predictions and experiments. This said, we have identified a pre-print under review, which used a colon epithelial carcinoma cell line as a model system to study impact of SARS-CoV-2 infection on cell physiology (Bojkova et al, 2020). In brief, this study has found that cholesterol synthesis is down-regulated, whereas synthesis of RNA-modifier proteins as well as carbon metabolism is up-regulated in infected cells. Furthermore, this study experimentally shown that inhibiting glycolysis as a whole with a drug decreases the replication rate of the SARS-CoV-2 in this model system. These findings, and additional findings from other cell lines and virus infections (El-Bacha et al, 2004; Munger et al, 2006) show that our overall findings are experimentally supported and that targets such as ENO, GAPD, PGM, and PGK, which involve in glycolysis and the entrance to the TCA cycle can be indeed promising drug targets for inhibiting SARS-CoV-2 replication in cells.

In terms of model result dependence on the model structure, we used the RECON2.2 model (Thiele et al, 2013; Swainston et al, 2016). This curated human genome scale model contains confidence levels for most included reactions and associated gene information, which can be improved by future studies updating the model structure (making use, e.g., of the more recent RECON3D [Brunk et al, 2018]). This is an area of active development for genome-scale models (Chindelevitch et al, 2015) and any future developments of model structure are bound to improve subsequent downstream analyses like this one. By adapting this model's biomass function to human lung cells—using available data from the Human Atlas Proteome project—we aimed to create a model that mimics the natural target cells of the SARS-CoV-2, namely lung and intestinal cells (Uhlén et al, 2015).

In our view, this approach should be more informative than using existing cell-specific models that are not experimentally shown the be SARS-CoV-2 targets, such as the macrophage model, which has been used recently in a pre-print to study SARS-CoV-2 infection (Renz et al, 2020). Besides using the human alveolar macrophage for the host model, that study has also used parsimonious hypotheses to build the virus biomass stoichiometry, assuming an arbitrary and single number for the copy number of each viral structural protein. Instead, and as explained above, we use an estimation based on experimental studies on coronaviruses, including SARS-CoV-2. As a result, some differences can be observed when comparing the two model's virus stoichiometry, mainly regarding the number of required ATP hydrolyses. These differences in host model and virus biomass construction are expected to lead

to different results between the two studies, yet, interestingly, both studies shared also some targets such as guanylate kinase (GK1).

The virus life cycle consists of environmental circulation, infection, and subsequent host cell attachment and entry, reproduction within the host cell, and exit for a new round of infection. The presented approach focuses solely on the reproduction within the host cell and the metabolic aspects of that. Although this is a limited focus, reproduction in the host cell is a crucial and essential aspect of the virus life cycle. The importance of this stage is highlighted in several studies, which demonstrate that viruses tend to re-program host metabolism for increased viral production (El-Bacha et al, 2004; Munger et al, 2006, 2008; Yu et al, 2011) or encode enzymes that can participate in host metabolic functions (Maynard et al, 2010a). These findings show that metabolic basis of host-virus interaction is crucial for the success of viruses and suggests that such interaction could be under significant evolutionary selection. Emergent viruses, such as SARS-CoV-2, are argued to not be well-adapted to their new host and undergo rapid evolution dictated by host-determined factors (Simmonds et al, 2019). It has been highlighted, for example, that there is a codon usage bias in virus genomes that possibly evolve in time to align with their host (Wong et al, 2010). The presented approach suggests that there might be a similar adaptation of viruses to their host metabolism.

We argue that differences in metabolic requirements of a virus versus its host could be a "physiological mismatch" that contributes to this evolutionary dynamic. Before metabolic adaptations happen, however, inhibition of the host metabolism might be a possible strategy to selectively inhibit reproduction of emergent viruses in new hosts. Specific host-based metabolic perturbations have already been shown experimentally to be effective against viruses (Munger et al, 2008; Carinhas et al, 2010), whereas general perturbation of end points of nucleotide biosynthesis through nucleoside analogs underpins the mode of action of several existing antiviral drugs. The predictions listed here present possible new antiviral targets that are primarily within central carbon metabolism, and in particular in glycolysis and oxidative phosphorylation. There are already several drugs that are shown to interact with the predicted enzymes in this study, opening up the possibility of experimentally testing the presented predictions using in vitro assays and cell cultures. In addition, latest techniques for controlling enzyme levels from genetic knockout to RNA-based interference strategies or optogenetic approaches (Sahin et al, 2007; Baaske et al, 2018) can be adapted to implement presented predictions involving flux enforcements.

In the development of host-based metabolic strategies to inhibit viruses, metabolic modelling, as presented here, can play a useful role. In particular, our approach can be applied relatively rapidly to any host–virus pair both for existing and emerging viruses, and allow generating experimentally testable hypotheses for virus inhibition. This approach can be applied as long as structural and genomic information can be converted into an estimation of biomass composition for the virus and a suitably detailed metabolic model for the host can be constructed. The former process can be enhanced by further databases of viral structural and genomic information, whereas the latter process would benefit from extending human-focussed efforts such as the human metabolic

atlas database (Robinson et al, 2020) to cover also cell lines of common animal hosts.

# Materials and Methods

### Human genome-scale metabolic model and its adaptation to human lung cells

To identify specific host metabolic reactions that can alter viral production, we make use of a generic human cell genome-scale metabolic model that has been previously developed (Thiele et al, 2013) and that has been subsequently curated and improved by the systems biology community (Swainston et al, 2016). This model, referred as RECON2.2 contains more than 8,000 reactions, many of which have associated gene and protein information (Swainston et al, 2016). This model also contains a pseudo reaction representing generic maintenance costs of a human cell, including ATP and precursor stoichiometries for proteins, DNA, RNA, lipids, and carbohydrates. This pseudo biomass maintenance reaction is primarily derived using information from human leukaemia cell lines (Thiele et al, 2013). This generic human cell model represents a consensus human metabolic capacity, and as such, its use in this study allows identification of widest range of possible metabolic reactions which can then be further scrutinized and sifted in a cell-specific context.

As discussed in the main text, when identifying metabolic perturbations that can selectively affect viral production without much affecting the host, the specifics of the used host maintenance representation must be cell specific. To align this representation to a lung cell, we have made use of the gene expression data available from the Human Protein Atlas project (Robinson et al, 2020). In particular, we have used the gene expression profile from mechanically homogenized whole tissue lung sample available from this project and available protein sequences from ENSEMBL database (Yates et al, 2020) to create a lung cell–specific stoichiometry for amino acids required for protein synthesis. The proportion of each amino acid in the composition of the new maintenance function was determined by converting the protein coding RNA's codons to amino acids, counting their frequency and weighing these with the normalised expression coefficient provided by the Human Protein Atlas project. The energy cost associated with amino acid polymerisation is also accounted for based on the length of the protein sequences and assuming 4.3 molecules of ATP hydrolysed to ADP per amino acid polymerisation (Quek et al, 2014). The stoichiometric coefficient for each amino acid in the new maintenance function is then scaled so as to represent the same weight as the original protein synthesis reaction in the RECON2.2 model (not accounting for ATP, ADP, Pi nor $H_2O$ in the scaling) (see the Data Availability section). The remaining elements of the maintenance function were retained as in the RECON2.2 model. The final, lung cell–specific maintenance function is provided in Table S3.

### Creation of SARS-CoV-2 virus biomass function

The biomass function for the SARS-CoV-2 is created as in a previous study (Aller et al, 2018) and by accounting for the composition and stoichiometry of proteins and genomic material in the virus.

Composition of protein and virus genome sequences are obtained from the National Centre for Biotechnology Information (NCBI) nucleotide database (accession number NC_045512). For one virus particle, the number of copies of genome ($C_g$) is assumed to be 1, whereas the number of copies of structural proteins is assumed to be 1,540 for membrane glycoproteins (M), 270 for surface glycoproteins (S), 490 for nucleocapsid phosphoproteins (N), and 5 for each of the other structural proteins of SARS-CoV-2. These protein copy numbers and stoichiometries are estimated from electron microscopy and mass spectrometry studies on other coronaviruses, including SARS-CoV (Neuman et al, 2006; 2011; 2008; Bárcena et al, 2009, Mahmoudabadi et al, 2017), and is further checked against a recent estimate specific to SARS-CoV-2 (Bar-On et al, 2020). The number of moles of each nucleotide required per virus particle is obtained from the sequence of the SARS-CoV-2 genome, accounting for the need to also produce an antisense strand. Similarly, the number of moles of each amino acid per virus particle is obtained by multiplying the amount of amino acid found in each protein sequence by the copy number assumed for that protein. Energetic costs in form of ATP stoichiometry is computed assuming 4.3 and 1.4 molecules of ATP is hydrolysed to ADP per amino acid and nucleotide polymerisation respectively (Quek et al, 2014). Once the total number of moles of each nucleotide, amino acid, and ATP required per virus particle is estimated, the total molar weight per virus particle is determined and the SARS-CoV-2 biomass biosynthesis reaction stoichiometry is expressed in mmol per gram of virus. The final biomass function is provided in Table S3, whereas the computer code used to calculate it is made available (see the Data Availability section). The modified version of the RECON2.2 containing the lung host biomass function described in the subsection above as well as the virus biomass function described in this section is available in SBML 3 version 1 format (Hucka et al, 2018) as Supplemental Data 1 and accessible online at BioModels database (Malik-Sheriff et al, 2020) under id MODEL2010280002.

### Simulation of the metabolic model

The integrated genome-scale metabolic model was simulated using the FBA approach (Bordbar et al, 2010). FBA assumes steady state of metabolic fluxes and implements linear optimisation to find one particular flux distribution across all reactions that can satisfy this assumption and that is optimal under given flux constraints and a certain optimality criterion. Here, we used the standard mathematical implementation of FBA as described before (Bordbar et al, 2014) and used maximisation of flux through the host maintenance or viral biomass pseudo reactions. All reaction flux constraints are kept as in the original RECON2.2 model except for extracellular transport reactions. The extracellular transport reactions are normally set to carry negative flux to represent uptake of metabolites from the media. In the RECON2.2 model all extracellular transport reactions' minimum flux values are set to –1,000 mmol $gDW^{-1}$ $h^{-1}$ (where DW stands for grams of dry weight) to represent a rich media (all exchange reactions allowed to carry flux). We have used here both this approach and additionally implemented a minimal media containing only essential metabolites, carbon and nitrogen source, and oxygen. The identification of the minimal media was achieved using a linear optimisation based

algorithmic approach (Senior et al, 2017), where a pseudo currency metabolite is added to all exchange reactions of the model and the flux for the extracellular transport reaction of this pseudo metabolite is systematically altered to identify a minimal set of exchange reactions that can still result in model optimisation. To implement the minimal media, the identified extracellular transport reactions' minimum flux values were set to −1,000 or to −10 mmol gDW$^{-1}$ h$^{-1}$, with all other extracellular transport reactions' minimum flux set to zero. The identified media composition is provided as Table S7 and a computational implementation of the described minimal media identification approach is provided in Python (see the Data Availability section).

## Data Availability

All relevant data are presented in the main and supplementary texts. The SBML model of the SARS-CoV-2–infected lung cell is also available at BioModels (Malik-Sheriff et al, 2020) under id; MODEL2010280002 (https://www.ebi.ac.uk/biomodels/MODEL2010280002). All source code used in this work are available on our research group GitHub web pages at: https://github.com/OSS-Lab/FBAhv.

## Supplementary Information

## Acknowledgements

We thank three anonymous reviewers for their insightful comments during the review process and members of the Soyer group for insightful discussions on model development and analysis. We acknowledge the support of our families, who have endured us during this computational study conducted under COVID-19–enforced lock down and home-working conditions. Funding statement: This work was funded by the University of Warwick and by the Biotechnological and Biological Sciences Research Council (BBSRC), with grants BB/T010150/1 (to OS Soyer) and BB/S506783/1 (to the University of Warwick).

### Author Contributions

H Delattre: software, formal analysis, validation, investigation, visualization, methodology, and writing—original draft, review, and editing.
K Sasidharan: resources, software, formal analysis, and writing—review and editing.
OS Soyer: conceptualization, software, formal analysis, supervision, funding acquisition, investigation, and writing—original draft, review, and editing.

### Conflict of Interest Statement

The authors declare that they have no conflict of interest.

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
