## [Reviewer comments · Life Science Alliance]

Life Science Alliance

Inhibiting the reproduction SARS-CoV-2 through perturbations in human cell metabolic network.

Hadrien Delattre, Kalesh Sasidharan, and Orkun Soyer

DOI: <https://doi.org/10.26508/lsa.202000869>

Corresponding author(s): Orkun Soyer, University of Warwick

Review Timeline:	Submission Date:	2020-08-05
	Editorial Decision:	2020-09-03
	Revision Received:	2020-10-02
	Editorial Decision:	2020-10-23
	Revision Received:	2020-11-02
	Accepted:	2020-11-11

Scientific Editor: Shachi Bhatt

Transaction Report:

September 3, 2020

Re: Life Science Alliance manuscript #LSA-2020-00869-T

Prof Orkun S Soyer
University of Warwick
Gibbet Hill Campus
Coventry CV4 7AL
United Kingdom

Dear Dr. Soyer,

Thank you for submitting your manuscript entitled "Inhibiting the reproduction of COVID-19-causing SARS-CoV-2 through perturbations in human cell metabolic network." to Life Science Alliance (LSA). The manuscript has been reviewed by the editors and outside referees (reviewer comments below). Although the topic was certainly of interest, serious technical and conceptual concerns voiced by the referees unfortunately preclude publication of the current version of the manuscript in LSA. We would be willing to consider a revised manuscript only if the revision addresses all the concerns of the reviewers 2 and 3 and the concerns from reviewer 1 about the practicality of enforcing a specific numerical range of flux via the two enzymes are addressed. We also encourage you to provide experimental evidence to robustly support the presented hypothesis, however these would not be required for publication. Please let us know if you are able to address the referees' comments and wish to submit a revised manuscript to LSA.

We would be happy to discuss the individual revision points further with you should this be helpful. While you are revising your manuscript, please also attend to the below editorial points to help expedite the publication of your manuscript. Please direct any editorial questions to the journal office. When submitting the revision, please include a letter addressing the reviewers' comments point by point.

We hope that the comments below will prove constructive as your work progresses. Thank you for this interesting contribution to Life Science Alliance. We are looking forward to receiving your revised manuscript.

Sincerely,

Shachi Bhatt, Ph.D.
Executive Editor
Life Science Alliance

B. MANUSCRIPT ORGANIZATION AND FORMATTING:

Reviewer #1 (Comments to the Authors (Required)):

Review of Delattre et al.

This is a timely topic. There is an urgent need to develop methods for identifying antiviral targets in general and specifically for the novel coronavirus. The authors use a genome scale model of human metabolism, Recon 2.2 by Thiele et al., to computationally investigate the impact of enzyme activity modulations (e.g., single and double knockouts) on the replication of SARS-Cov-2 in lung cells. The authors largely use the published model, with two modifications. One modification is to add a reaction representing the production of SARS-Cov-2 biomass. The second modification is to replace the host cell maintenance reaction of Recon 2.2 with a pseudo reaction that more specifically models the lung based on gene expression data from the Human Protein Atlas project. The authors use a standard flux balance analysis (FBA) approach to calculate reaction fluxes under 'minimal' and 'rich' media conditions where only essential and all nutrients are available for uptake

by the cells.

The key result presented in the manuscript is that for some reactions, there are flux values that are compatible with optimal host cell maintenance, but incompatible with optimal virus production. Specifically, the authors report that enforcing the flux through threonine deaminase (THRD) to lie between 8.94 and 9.14 mmol/gDW/h, combined with enforcing the flux through another one of four selected reactions (three from the TCA cycle and one from purine metabolism) could reduce viral biosynthesis by 17% compared to the optimal rate.

Overall, I think the approach has merit. The virus depends on the host cell to synthesize its building blocks, and thus competes with the host for energy and biosynthetic precursors. In principle, selectively inhibiting key host enzymes could reduce the availability of biosynthetic precursors for the virus and thus attenuate its rate of replication in the host cell. On other hand, these enzymes also impact the host cell (as the authors report).

These are intuitive conclusions, and it is unclear that the authors provide concrete evidence of novel enzymatic targets that could be practically manipulated to achieve the stated aim of attenuating virus production without harming the host cell's viability. It is well known that biosynthetic precursors depend on central carbon metabolism, and thus it is not surprising that inhibiting enzymes in these pathways would impact both virus production and host cell maintenance. Given that sub-classes of amino acids and nucleotides share common precursors that derive from central carbon metabolism, it is also not surprising that the reactions required to produce the virus' components and maintain the host cell largely overlap. Enforcing the flux through an enzyme to a specific numerical range appears implausible, and achieving this for two enzymes is even more difficult. The authors list several compounds that are currently in use as drugs or dietary supplements, but it is unclear how these molecules would be used to enforce enzymatic fluxes to specific ranges, let alone avoid off-target effects. Importantly, there is no experimental validation of the authors' predictions, even for single knockouts.

Additional comments.

1. The lung is not a homogeneous tissue. Which cell type(s) are the authors modeling? Also, the methods text refers to gene expression, rather than protein expression. Protein abundance may not correlate with gene expression.
2. It is unclear why an FBA approach is needed to compare the nucleic acid and amino acid requirements of the virus and host cell. This could be done comparing the stoichiometries of the two pseudo reactions.
3. The rationale for determining that a rich medium state is less physiologically relevant than a minimal medium state is unclear. Further, if this is the case, then it would appear that analyzing the rich medium state was unnecessary.
4. Threonine, leucine, and isoleucine are all essential amino acids. Rather than enforcing their flux to specific ranges, have the authors investigated the effect of restricting their availability in the medium?
5. The choice of host maintenance as the objective function should be further explained. Typically, FBA objectives are validated by comparing the calculated and experimentally observed metabolic behavior. In the infected cell, which objective (host vs. virus) would dominate? Are there prior studies on other viruses that could provide insights into the impact of viral infection on host cell

metabolic objectives?

Reviewer #2 (Comments to the Authors (Required)):

Understanding the interrelationship between SARS-CoV-2 and human host will not only establish targets for drug development but will also help to reveal the pathways related to the host's response to COVID-19 infection. Delattre et al. employed a systems biology approach by utilizing the cell specific GEM of human lung cell with structural information of SARS-CoV-2 to study host-pathogen interactions. The authors generated genome-scale metabolic model (GEM) of a human lung cell infected with SARS-CoV-2 using RECON2.2.

The authors did make some efforts and I think the purpose of the study is quite interesting, however I could not see novelty here except the implementation a pseudo-reaction to the published GEM and performing synthetic knock-out analysis. This work can be a potentially publishable paper if the authors meet the fundamental issues listed below.

1. Authors provided several key data (e.g. virus biomass function) only in the supplementary material. However, no supplementary files are provided. Therefore, reviewing is done with only main text and figures.
2. In material methods section there are excessive details about well-known studies/methodologies (e.g. RECON 2.2, FBA). However, the manuscript does not explain important parameter choices that applied here. Especially creation of SARS-CoV-2 biomass function is very poorly written. The computational approach used in this study was developed by Aller et al. (2018) and applied with various epidemic viruses in the same manuscript. Authors must rewrite the methodology section clearly.
3. Flux distributions are calculated by using FBA as stated line 378. However, in FBA, multiple flux states can achieve the same optimum. Other simulation approaches should be tried to obtain a comprehensive picture of flux changes.
4. It is unclear why authors selected "the 80% or less" reduction of virus optima of the original. Is there any support if the rates have any significance in terms of reported cell growth/virus replication?
5. In results and discussion section, there is no/little discussion, only findings are given.

Minor issues:

1. Lines 219-313 in conclusion more fits to be in introduction section.
2. In the title instead of using "COVID-19-causing SARS-CoV-2" authors might reconsider writing as "COVID-19-causing coronavirus" or only "SARS-CoV-2" would be enough.
3. The link for code availability is not working.
4. Affiliation list is incomplete.

Reviewer #3 (Comments to the Authors (Required)):

Summary

=====

In this manuscript, the authors describe their computational approach to fight the ongoing COVID-19 pandemic. To this end, they have created a genome-scale metabolic model based on RECON 2.2 (Thiele 2003; Swainston 2016). They adjusted this model to human lung cells using gene-expression data from the Human Protein Atlas project (Robinson 2020). Next, the authors integrated a viral biomass objective function into their tissue-specific model based on the method by Aller et al. (2018). Hence, the model reflects the situation that the virus has already reprogrammed the cell and tries to replicate at a maximal rate. By applying a minimal media composition assumption, the authors conducted a flux variability analysis and systematically assessed if reaction knockouts can inhibit viral growth while sustaining regular cell maintenance. From this analysis, the authors conclude that existing drugs could be repurposed to inhibit those biochemical reactions with a high impact on viral reproduction and simultaneously small disturbance of lung-cell maintenance. The authors present their key findings in a neatly arranged table to support follow-up experimental studies.

Evaluation

=====

The manuscript is generally well written and also summarizes very recent work on the probably most pressing topic of humankind at this time. The paper is highly timely, and if the predictions are correct, it could have a high impact on the development of a treatment strategy against this ongoing pandemic. However, the approach follows in large parts very closely the proposed method by Renz et al. (2020), <https://doi.org/10.5281/zenodo.3752640>. The main difference seems to be that the manuscript at hand is based on Recon 2.2. In contrast, Renz et al. built their integrated host-virus model on the human alveolar macrophage model by Bordbar et al. (2010) that the authors also cite in this submission. The main goal right now is to find ways to treat the pandemic, and any step towards new medication is of great importance. This manuscript at hand is based on a different model than the earlier approach, which allows the authors to check if results can be confirmed in different ways.

Major Points

=====

Because of the substantial overlap in approach and methodology, it would be essential that the authors compare their work to the work of Renz et al., which is an earlier approach. In particular, it would be of high interest if Renz et al. found that the human guanylate kinase 1 (GK1) is among the promising targets when building upon a tissue-specific model from Recon 2.2. The authors should also use Renz et al.'s work to compare if their viral biomass objective functions of both models diverge. If so, please explain and clarify what the difference and their origins are.

The authors mention several supplementary files in their text, which is nice. Unfortunately, these were not accessible for the review process. It is, therefore, not possible to fully assess the quality of their work. For the re-review, please make sure to upload all supplementary files. The link quoted in the manuscript ("OSS research group GitHub web pages") does not work; it is highlighted in blue

and underscored, but not clickable.

For the model, it would be best to encode it in SBML format and to upload this model directly to the BioModels Database. When doing so, please request reviewer access and write the login details into the draft manuscript. By doing so, reviewers will be able to download the model and explore it with a software of choice. In this way, the authors ensure the reproducibility of their work. In particular, for the current matter, reproducibility is of the highest importance to developing a treatment.

Minor Points

=====

There is no reason for spelling out Greek letters such as "alpha" or "gamma." Please write correctly using the corresponding symbols.

Figure 3 comes with tiny axes labels. Please increase the font.

Please make sure to always add a comma after "i.e." (and, by the way, also after "e.g.").

Reviewer 1

This is a timely topic. There is an urgent need to develop methods for identifying antiviral targets in general and specifically for the novel coronavirus. The authors use a genome scale model of human metabolism, Recon 2.2 by Thiele et al., to computationally investigate the impact of enzyme activity modulations (e.g., single and double knockouts) on the replication of SARS-Cov-2 in lung cells. The authors largely use the published model, with two modifications. One modification is to add a reaction representing the production of SARS-Cov-2 biomass. The second modification is to replace the host cell maintenance reaction of Recon 2.2 with a pseudo reaction that more specifically models the lung based on gene expression data from the Human Protein Atlas project. The authors use a standard flux balance analysis (FBA) approach to calculate reaction fluxes under 'minimal' and 'rich' media conditions where only essential and all nutrients are available for uptake by the cells.

The key result presented in the manuscript is that for some reactions, there are flux values that are compatible with optimal host cell maintenance, but incompatible with optimal virus production. Specifically, the authors report that enforcing the flux through threonine deaminase (THRD) to lie between 8.94 and 9.14 mmol/gDW/h, combined with enforcing the flux through another one of four selected reactions (three from the TCA cycle and one from purine metabolism) could reduce viral biosynthesis by 17% compared to the optimal rate.

Overall, I think the approach has merit. The virus depends on the host cell to synthesize its building blocks, and thus competes with the host for energy and biosynthetic precursors. In principle, selectively inhibiting key host enzymes could reduce the availability of biosynthetic precursors for the virus and thus attenuate its rate of replication in the host cell. On other hand, these enzymes also impact the host cell (as the authors report).

We thank the reviewer for this overall summary, which is largely accurate. We note that the presented approach identifies several possible alterations in the host metabolism, and not just the one highlighted by the reviewer above. As discussed in the manuscript, some of these are found to affect both host and virus, while some only the virus.

These are intuitive conclusions, and it is unclear that the authors provide concrete evidence of novel enzymatic targets that could be practically manipulated to achieve the stated aim of attenuating virus production without harming the host cell's viability. It is well known that biosynthetic precursors depend on central carbon metabolism, and thus it is not surprising that inhibiting enzymes in these pathways would impact both virus production and host cell maintenance. Given that sub-classes of amino acids and nucleotides share common precursors that derive from central carbon metabolism, it is also not surprising that the reactions required to produce the virus' components and maintain the host cell largely overlap. Enforcing the flux through an enzyme to a specific numerical range appears implausible, and achieving this for two enzymes is even more difficult. The authors list several compounds that are currently in use as drugs or dietary supplements, but it is unclear how these molecules would be used to enforce enzymatic fluxes to specific ranges, let alone avoid off-target effects. Importantly, there is no experimental validation of the authors' predictions, even for single knockouts.

We have tried to provide support from experimental studies for some of our predictions, however, a full experimental test lies behind the scope of this work. We have extended our discussion about possible experimental implementations of the predictions. Predictions that are found to affect both host and virus can be readily implemented as knockouts and could still be relevant. Predictions that affect only the virus require fine tuning of enzyme levels, which can be achieved with RNA interference methods.

We have re-emphasized these points in the revised manuscript, as well as including a related sentence in the revised *Abstract*.

1. The lung is not a homogeneous tissue. Which cell type(s) are the authors modeling? Also, the methods text refers to gene expression, rather than protein expression. Protein abundance may not correlate with gene expression.

The data we use comes from the Human Atlas Proteome project, the transcriptomic measurements were done from mechanically homogenized whole tissue samples (see doi: [10.1074/mcp.M113.035600](https://doi.org/10.1074/mcp.M113.035600)). The Human Atlas project has also performed comparisons between tissue-based and cell line – based transcriptomic profiles and found that cell-based data differs from the tissue-based one – it has been argued that the tissue-based data is more representative of the *in vivo* conditions (doi: [10.1074/mcp.M113.035600](https://doi.org/10.1074/mcp.M113.035600)). We agree with the reviewer that this information is important in the understanding of the outcomes of our model, thus, we added a relevant section to the revised *Materials and Methods*.

Results produced by the Human Atlas project have shown that their transcriptomic data is generally well correlated to protein abundance (doi:[10.1038/nature13319](https://doi.org/10.1038/nature13319); doi: [10.1038/msb.2010.106](https://doi.org/10.1038/msb.2010.106)).

2. It is unclear why an FBA approach is needed to compare the nucleic acid and amino acid requirements of the virus and host cell. This could be done comparing the stoichiometries of the two pseudo reactions.

We have indeed compared the stoichiometries of the two pseudo reactions to reveal the differentially used amino acids and nucleotides in the host vs. virus (see Figure 3). The FBA approach goes beyond this and identifies possible enzyme/reaction perturbations that can affect host or virus biomass either in a similar manner or differentially. This latter information is not trivial to guess from the biomass reaction stoichiometry alone, due to the branched nature of cell metabolism.

3. The rationale for determining that a rich medium state is less physiologically relevant than a minimal medium state is unclear. Further, if this is the case, then it would appear that analyzing the rich medium state was unnecessary.

What we call ‘rich medium’ is a model where all transport reactions existing in the RECON2.2. are allowed to carry flux. Thus, this represents a situation where a human cell has access to all nutrients that it has transporters for. We believe that such a situation would rarely, if at all, be achieved *in vivo*.

We have opted to still keep the rich medium results, as we believe this provides a reference point. Any reaction affecting host or biomass under the rich medium case is likely to have an effect under more restrictive medium conditions too. Also, having the rich medium model can be relevant for future studies, where experimental conditions can achieve such a medium.

4. Threonine, leucine, and isoleucine are all essential amino acids. Rather than

enforcing their flux to specific ranges, have the authors investigated the effect of restricting their availability in the medium?

We have now added this type of media-restriction analysis for each component of the minimal medium we used in the manuscript. Results are compiled as a new supplementary file (see *Supplementary material 7*). In brief, we find that limiting the availability of threonine, glyclphenylalanine or 3-methyl-2-oxovalerate in the medium decreases the value of the virus optimum function more readily compared to effects of these limitations on the host biomass function. This new result has been added and discussed in the revised manuscript as a new paragraph.

5. The choice of host maintenance as the objective function should be further explained. Typically, FBA objectives are validated by comparing the calculated and experimentally observed metabolic behavior. In the infected cell, which objective (host vs. virus) would dominate? Are there prior studies on other viruses that could provide insights into the impact of viral infection on host cell metabolic objectives?

We have now added a discussion of model predictions in light of experimental data in the *Results&Discussion* section in the revised manuscript.

There is unfortunately not much available literature yet on COVID-19 infected cells and their metabolism, to allow us detailed comparison between model predictions and experiments. This said, we have identified a pre-print under review, which used a colon epithelial carcinoma cell line (Caco-212) as a model system to study impact of SARS-CoV-2 infection on cell physiology. In brief, this study has found that cholesterol synthesis is downregulated, while synthesis of RNA-modifier proteins, as well as carbon-metabolism are upregulated in infected cells. Furthermore, this study experimentally shown that inhibiting glycolysis as a whole with a drug decreases the replication rate of the SARS-CoV-2 in this model system (doi: 10.21203/rs.3.rs-17218/v1).

These findings, and additional findings from other cell-lines and virus infections (see citations 8, 12, 23, and 24 in the main text) show that our overall findings are experimentally supported and that targets such as ENO, GAPD, PGM and PGK, which involve in glycolysis and the entrance to the TCA cycle can be indeed promising drug-targets for inhibiting SARS-CoV-2 replication in cells.

Reviewer 2

Understanding the interrelationship between SARS-CoV-2 and human host will not only establish targets for drug development but will also help to reveal the pathways related to the host's response to COVID-19 infection. Delattre et al. employed a systems biology approach by utilizing the cell specific GEM of human lung cell with structural information of SARS-CoV-2 to study host-pathogen interactions. The authors generated genome-scale metabolic model (GEM) of a human lung cell infected with SARS-CoV-2 using RECON2.2.

The authors did make some efforts and I think the purpose of the study is quite interesting, however I could not see novelty here except the implementation a pseudo-reaction to the published GEM and performing synthetic knock-out analysis. This work can be a potentially publishable paper if the authors meet the fundamental issues listed below.

1. Authors provided several key data (e.g. virus biomass function) only in the supplementary material. However, no supplementary files are provided. Therefore, reviewing is done with only main text and figures.

We apologise for this inconvenience. We have included supplementary files in the original submission, but these must have been lost during (automated) transfer between journals. We have made sure now to include all supplementary material.

2. In material methods section there are excessive details about well-known studies/methodologies (e.g. RECON 2.2, FBA). However, the manuscript does not explain important parameter choices that applied here. Especially creation of SARS-CoV-2 biomass function is very poorly written. The computational approach used in this study was developed by Aller et al. (2018) and applied with various epidemic viruses in the same manuscript. Authors must rewrite the methodology section clearly.

We have now included more details about the virus biomass creation on lines 418-439 of the revised manuscript.

3. Flux distributions are calculated by using FBA as stated line 378. However, in FBA, multiple flux states can achieve the same optimum. Other simulation approaches should be tried to obtain a comprehensive picture of flux changes.

The only other approach we can think of, for studying different and equally optimal flux solutions, could be an analysis of the elementary flux modes in the model (doi: 10.1007/s002850200143). While methods exist to make it feasible for a genome-scale model, they remain computationally onerous approaches (doi: 10.1093/bioinformatics/btu021), and we would have to perform it on all model variants we study here (i.e. the knockouts and double knockouts), resulting in over 8000^2 models. Thus, we do not see any alternative approach possible here at this point.

We would also like to highlight that our results under different conditions (e.g. different media) lead to the same perturbation effects of knockouts in general. Thus, we do not expect that flux solutions that are equally optimal to give us different insights on perturbation effects.

4. It is unclear why authors selected "the 80% or less" reduction of virus optima of the original. Is there any support if the rates have any significance in terms of reported cell growth/virus replication?

We used this as an arbitrary cut-off to highlight key findings. We have included all simulations results below that threshold in the supplementary materials

5. In results and discussion section, there is no/little discussion, only findings are given.

We have now extended the discussion in this section, particularly focusing on possible experimental implementation of the predictions and also their evaluation against any available experimental data.

Minor issues:

1. Lines 219-313 in conclusion more fits to be in introduction section.

We have moved these lines as suggested by the reviewer.

2. In the title instead of using "COVID-19-causing SARS-CoV-2" authors might reconsider writing as "COVID-19-causing coronavirus" or only "SARS-CoV-2" would be enough.

We agree with the reviewer. We have opted for using “SARS-CoV-2” only.

3. *The link for code availability is not working.*

We apologise for this error – this is now fixed.

4. *Affiliation list is incomplete.*

We corrected this.

Reviewer 3

In this manuscript, the authors describe their computational approach to fight the ongoing COVID-19 pandemic. To this end, they have created a genome-scale metabolic model based on RECON 2.2 (Thiele 203; Swainston 2016). They adjusted this model to human lung cells using gene-expression data from the Human Protein Atlas project (Robinson 2020). Next, the authors integrated a viral biomass objective function into their tissue-specific model based on the method by Aller et al. (2018). Hence, the model reflects the situation that the virus has already reprogrammed the cell and tries to replicate at a maximal rate. By applying a minimal media composition assumption, the authors conducted a flux variability analysis and systematically assessed if reaction knockouts can inhibit viral growth while sustaining regular cell maintenance. From this analysis, the authors conclude that existing drugs could be repurposed to inhibit those biochemical reactions with a high impact on viral reproduction and simultaneously small disturbance of lung-cell maintenance. The authors present their key findings in a neatly arranged table to support follow-up experimental studies. We thank the reviewer for this overall summary, which is largely accurate.

The manuscript is generally well written and also summarizes very recent work on the probably most pressing topic of humankind at this time. The paper is highly timely, and if the predictions are correct, it could have a high impact on the development of a treatment strategy against this ongoing pandemic. However, the approach follows in large parts very closely the proposed method by Renz et al. (2020), <https://doi.org/10.5281/zenodo.3752640>. The main difference seems to be that the manuscript at hand is based on Recon 2.2. In contrast, Renz et al. built their integrated host-virus model on the human alveolar macrophage model by Bordbar et al. (2010) that the authors also cite in this submission. The main goal right now is to find ways to treat the pandemic, and any step towards new medication is of great importance. This manuscript at hand is based on a different model than the earlier approach, which allows the authors to check if results can be confirmed in different ways.

*We thank the reviewer for this relevant pre-print, which we were not aware of. We are now including a citation to this pre-print and briefly discuss it in the revised *Conclusion* section.*

Because of the substantial overlap in approach and methodology, it would be essential that the authors compare their work to the work of Renz et al., which is an earlier approach. In particular, it would be of high interest if Renz et al. found that the human guanylate kinase 1 (GK1) is among the promising targets when building upon a tissue-specific model from Recon 2.2.

The authors should also use Renz et al.'s work to compare if their viral biomass objective functions of both models diverge. If so, please explain and clarify what the difference and their origins are.

While we note that the Renz et. al. study is also a pre-print and has not been through peer review yet, we agree that it is important and useful to compare the two studies in terms of their general features. We have now done so in the revised manuscript. In brief, the Renz study used the published human macrophage model as their ‘host’ system, which is not necessarily a target of SARS-CoV-2. Instead, in the presented study, we have used the broad RECON2.2. model and adapted it to lung tissue through the use of lung-tissue based expression and proteomics data from Human Atlas Project. This was done to account for the fact that the main infection target of SARS-CoV-2 seems to be human lung cells (along with intestinal tissue, as we discuss in the manuscript). Moreover, regarding the method to determine the virus biomass function, both Renz et al and our study used the method proposed by Aller et al (doi: 10.1098/rsif.2018.0125). However, there are differences in the hypotheses used to determine the stoichiometry of the virus biomass reaction; while Renz et al assumed the same single number (between 200 and 1200) for the copy number of all the structural viral proteins, we used different numbers for each of the different structural viral proteins based on electron microscopy and mass spectrometry studies on other coronaviruses, including SARS-CoV (see main text). We also consider different ATP costs for DNA and protein polymerization, while Renz et al considered ATP costs of these processes to be the same. We cannot provide a more detailed comparison between our virus biomass function and the one used by Renz et al since we weren’t able to find its formula in their manuscript. Hence, the results between the models are expected to be different, due to differences in the host model structure, and cannot be directly comparable.

Despite such difference in models used, the GK1 perturbation (when considered as part of a double knockout) was among the key results in our study as well (see supplementary File S2).

The authors mention several supplementary files in their text, which is nice. Unfortunately, these were not accessible for the review process. It is, therefore, not possible to fully assess the quality of their work. For the re-review, please make sure to upload all supplementary files. The link quoted in the manuscript ("OSS research group GitHub web pages") does not work; it is highlighted in blue and underscored, but not clickable.

We apologise for this inconvenience. We have surely included supplementary files in the original submission, but these must have been lost during (automated) transfer between journals. The link worked in our submitted docx files, but might have been broken during document conversion to pdf. It is now included as the full link, which is: <https://github.com/OSS-Lab>. We have also made sure now to include all supplementary material.

For the model, it would be best to encode it in SBML format and to upload this model directly to the BioModels Database. When doing so, please request reviewer access and write the login details into the draft manuscript. By doing so, reviewers will be able to download the model and explore it with a software of choice. In this way, the authors ensure the reproducibility of their work. In particular, for the current matter, reproducibility is of the highest importance to developing a treatment.

We have now included SBML versions of the model as *Supplementary file 8*.

Minor Points

=====

There is no reason for spelling out Greek letters such as "alpha" or "gamma." Please write correctly using the corresponding symbols.

Corrected.

Figure 3 comes with tiny axes labels. Please increase the font.

Corrected.

Please make sure to always add a comma after "i.e." (and, by the way, also after "e.g.").

Corrected.

October 23, 2020

RE: Life Science Alliance Manuscript #LSA-2020-00869-TR

Prof. Orkun S Soyer
University of Warwick
Gibbet Hill Campus
Coventry CV4 7AL
United Kingdom

Dear Dr. Soyer,

Thank you for submitting your revised manuscript entitled "Inhibiting the reproduction SARS-CoV-2 through perturbations in human cell metabolic network.". We would be happy to publish your paper in Life Science Alliance pending revisions to address a number of (addressable) concerns from Reviewer 3 and final revisions necessary to meet our formatting guidelines.

Along with the points listed below, please also attend to the following:

- please add a conflict of interest statement to the main manuscript text
- please add your figure legends to your main manuscript text (including your main figures, supplementary figures, and table legends)
- please use the [10 author names, et al.] format in your references (i.e. limit the author names to the first 10)
- please add a callout for Figure 2A in your main manuscript text
- please make sure the manuscript sections and section order match with LSA's guidelines (<https://www.life-science-alliance.org/manuscript-prep#format>) - for eg. please separate the Results & Discussion section into 2 separate - 1 Results and 1 Discussion section
- please also deposit the model in the Biomodels Database and include the accession information in the Data Availability section (you can couple the code availability with this section too)
- The supplemental material file format, as it is right now, is difficult to read, would it be helpful to put it in a tabular format instead?

A. FINAL FILES:

B. MANUSCRIPT ORGANIZATION AND FORMATTING:

Sincerely,

Shachi Bhatt, Ph.D.
Executive Editor

Reviewer #2 (Comments to the Authors (Required)):

The authors did an excellent work during the revision. I recommend the publication of the paper in its current form.

Reviewer #3 (Comments to the Authors (Required)):

Thanks to the authors for implementing many improvements to their manuscript. A few aspects still need clarification before it can be published.

1) Their SBML file is not valid. The official validator at <http://sbml.org/Facilities/Validator/> displays four error messages. These errors require correction before publication. As much as possible, warnings should also be fixed.

2) Recon3D (<https://dx.doi.org/10.1038/nbt.4072>) is a significantly updated human cell model compared to Recon 2.2 (<https://doi.org/10.1007/s11306-016-1051>, which the authors used in their work. Could they please briefly motivate their choice of working with a previous version?

3) The name of the model is still "Swainston2016 - Reconstruction of human metabolic network (Recon 2.2)," and the model identifier is "MODEL1603150001," but the authors have created a modified model that includes the virus. It seems more appropriate to change the identifier and name of the model. Also, maybe the authors can indicate which reaction in their model refers to viral biomass production.

4) The authors nicely summarized the comparison of their approach to that in the preprint of Renz et al. (<https://doi.org/10.5281/zenodo.3752640>), particularly regarding the viral biomass objective function, which is the essential aspect. However, it is important to note that Renz et al. state in their article that they made their model accessible at <https://identifiers.org/biomodels.db/MODEL2003020001>. This SBML file contains the viral biomass function (VBOF) as well as the host maintenance function. It would still be favorable if the authors could use the SBML model to compare the differences they could not compare based on the manuscript alone. The desired table for the VBOF function can be generated from the SBML file relatively quickly, e.g., using a combination of libSBML's Python interface (<https://doi.org/10.1093/bioinformatics/btn051>) and the Pandas package.

5) Following the FAIR principles (Findable, Accessible, Interoperable, and Reproducible, <https://doi.org/10.1038/sdata.2016.18>), it is also highly important that the authors make their model not only available as a supplement to this article. Instead, they should deposit it in BioModels Database (<https://doi.org/10.1093/nar/gkz1055>) at <http://biomodels.net>. BioModels has created a special section for COVID-19 related models at <http://www.ebi.ac.uk/biomodels/covid-19>. It would be of high importance to also have this contribution there. There are multiple reasons why uploading the model to BioModels is much more favorable than including it in the supplements of an article. First, if a model is stored in BioModels, it is possible to have revisions attached to it, which is not the

case for a static and final model supplement to an article. No revision can be made to supplements. Next, the BioModels team of professional curators is continuously improving all models, checking the integrity, validity, annotation, and many additional features. This does not happen to supplements of an article. Moreover, the authors themselves did not check the validity of the model. Also, BioModels collects computational models in one place. It is a searchable infrastructure in which modelers can find models by keywords. Finding an SBML file that is distributed as a supplement to an individual article is much more difficult. In other words, supplement eight should be removed from the article again, and instead, the link (via identifiers.org) should be included in the manuscript in a section "Availability."

6) Please cite used resources using the latest publication, such as SBML (<https://doi.org/10.15252/msb.20199110>). Possibly, the specification of SBML Level 3 Version 1 could also be cited (<https://doi.org/10.1515/jib-2017-0080>). As it should be known, SBML as a resource requires such citations to secure ongoing funding and deserves general support by its users since it is essential to many efforts in systems biology.

November 11, 2020

RE: Life Science Alliance Manuscript #LSA-2020-00869-TRR

Prof. Orkun S Soyer
University of Warwick
Gibbet Hill Campus
Coventry CV4 7AL
United Kingdom

Dear Dr. Soyer,

Thank you for submitting your Research Article entitled "Inhibiting the reproduction SARS-CoV-2 through perturbations in human cell metabolic network.". It is a pleasure to let you know that your manuscript is now accepted for publication in Life Science Alliance. Congratulations on this interesting work.

DISTRIBUTION OF MATERIALS:

Again, congratulations on a very nice paper. I hope you found the review process to be constructive and are pleased with how the manuscript was handled editorially. We look forward to future exciting submissions from your lab.

Sincerely,

Shachi Bhatt, Ph.D.

Executive Editor

Life Science Alliance

<https://www.lsjournal.org/>
